# COVID-19 and Pulmonary Hypertension in Children: What Do We Know So Far?

**DOI:** 10.3390/medicina56120716

**Published:** 2020-12-19

**Authors:** Bibhuti B Das

**Affiliations:** Department of Pediatric, Division of Pediatric Cardiology, Baylor College of Medicine, Texas Children’s Hospital Austin Specialty Care, Austin, TX 78759, USA; bbdas@texaschildrens.org

**Keywords:** SARS-CoV-2, COVID-19, pediatric pulmonary hypertension

## Abstract

The interplay between coronavirus disease 2019 (COVID-19) and pulmonary hypertension (PH) in children is unknown. Adults with PH are at potential risk for severe complications and high mortality due to associated comorbidities. It is difficult to extrapolate the outcomes of COVID-19 in adults to pediatric PH patients. Overall, a small number of COVID-19 cases is reported in patients with preexisting PH. Several factors may be responsible for the low incidence of COVID-19 in children with PH. Pulmonary hypertension is a rare disease, testing is not universal, and patients may have followed more rigorously the Center for Disease Control’s guidelines recommended for personal protection with mask-wearing, social distancing, and hand sanitization through ongoing health education. The small number of COVID-19 cases in patients with preexisting PH does not support that PH is protective for COVID-19. However, medications used to treat PH may have some protection against COVID-19. This review discusses the pathophysiology of PH occurring with COVID-19, differences between children and adults with COVID-19, strategies for management of preexisting PH in children during the ongoing pandemic, and its impact within the field of PH.

## 1. Introduction

Coronavirus disease 2019 (COVID-19) has had a catastrophic impact on global health, with more than 60 million confirmed cases worldwide and more than 1.4 million deaths [1]. Children are increasingly infected with the Severe Acute Respiratory Syndrome Coronavirus 2 (SARS-CoV-2) as the pandemic unfolds. The most frequently reported underlying medical conditions in children with COVID-19 are chronic lung disease, obesity, neurodevelopmental disabilities, and congenital heart disease [2]. Patients with pulmonary hypertension (PH) are vulnerable to SARS-CoV-2 as it is primarily a respiratory virus. The incidence of COVID-19 in adults with PH is 2.1 cases per 1000 patients [3]. There is little data on the incidence of SARS-CoV-2 in children with preexisting PH, or the incidence of PH occurring in conjunction with COVID-19. A systematic search on electronic databases (PubMed, Scopus, medRxiv, and bioRxV) was performed using the terms “children with preexisting PH and SARS-CoV-2”, “PH in children occurring with COVID-19”, “impact of COVID-19 on preexisting PH”, “therapies for PH in COVID-19”, and “PH during COVID-19 pandemic”. There have only been two pre-existing severe PH cases without pulmonary thromboembolism with COVID-19 reported in children: a 16-year-old girl with mitral stenosis and severe PH; and a 6-month-old infant with severe PH and right ventricular (RV) failure [4,5]. One should be cautious when interpreting small case numbers of COVID-19 cases in children with preexisting PH for several reasons. Pulmonary hypertension is a rare disease. Testing for SARS-CoV-2 is not universal. Patients may have followed more rigorously the Center for Disease Control’s (CDC’s) recommended personal protection with mask-wearing, social distancing, and hand sanitization; and hypothetically, medications used to treat PH may have some protection against COVID-19. There is no evidence to support that PH could be protective for COVID-19, but individuals with PH do not appear to be at increased risk of contracting COVID-19 compared to the general public. The purpose of the review is to discuss the pathophysiology of PH occurring with COVID-19, differences between children and adults with COVID-19, strategies for managing PH in children during the pandemic, and the impact of COVID-19 within the field of PH.

## 2. Pathophysiology of PH Occurring with COVID-19

The novel coronavirus, SARS-CoV-2, produces its effects through its spike protein interaction with the Angiotensin-Converting Enzyme 2 (ACE2) receptors highly expressed in alveolar lung cells and the vascular endothelium in patients who have COVID-19 [6]. Viral cell entry requires the serine protease, transmembrane serine protease 2 (TMPRSS2), to be expressed on the host cell [7]. The virus–receptor interaction results in a deficiency of ACE2 at the cell membrane due to the virus’s internalization along with the receptor by endocytosis [8]. Deficiency/downregulation of ACE2, an enzyme that physiologically counters the renin-angiotensin-aldosterone (RAS) system, degrades angiotensin II and attenuates its effects on endothelial damage, vasoconstriction, and fibrosis [9]. On the other hand, there is a linear association between angiotensinogen II levels in plasma and viral load and the severity of lung injury in patients with COVID-19 [10]. Several putative mechanisms of lung injury and PH in COVID-19 disease are illustrated in Figure 1. Endothelial dysfunction is the key to initiate the cascade of events leading to ventilation/perfusion mismatch, hypoxia, vasoconstriction, and PH. Further increase in pulmonary artery pressure contributes to an increase in RV afterload, leading to RV dysfunction and heart failure.

In children, ACE2 expression and TMPRSS2 are decreased in the lungs [11], but it is not clear to what extent this plays a protective factor against SARS-CoV-2. The histopathology study of lungs in adult patients with COVID-19 has shown severe damage to endothelial cells, disruption of the endothelial barrier’s integrity, and extensive thrombosis [12]. A distinctive feature of COVID-19-related acute respiratory distress syndrome (ARDS) is vascular angiogenesis [13]. The extensive lung damage results in ARDS and subsequent interstitial fibrosis. Hypoxia due to ARDS results in pulmonary vasoconstriction and acute PH. Systemic inflammatory response due to cytokine storm and damage to the endothelium may result in hypercoagulable state and intravascular thrombosis, although the exact mechanism is not known. The hyperinflammatory response, fibrosis, and thrombosis further facilitate PH and subsequent right ventricle (RV) dysfunction. SARS-CoV-2 causes endothelial dysfunction, diffuse microangiopathy, and microthrombosis, leading to ventilation–perfusion mismatch; and intrapulmonary right-to-left shunting aggravates further hypoxia and a vicious cycle leading to remodeling of the pulmonary vasculature. The prognostic impact of COVID-19 in preexisting PH patients with already pulmonary vascular remodeling is unknown and needs future studies.

## 3. Differences between Children and Adults in COVID-19

There are significant differences between children and adults with incidence, severity, and complications of COVID-19. Adults with COVID-19 can present with myocarditis, myocardial infarction, cardiomyopathy, cardiogenic shock, acute cor-pulmonale, and arrhythmias, but these manifestations are relatively rare in pediatric patients [14]. The cardiac findings reported in children with COVID-19 are myocarditis, myocardial dysfunction, and coronary artery involvement [14]. In adults, preexisting conditions such as hypertension, diabetes, and obesity lead to poor outcomes with COVID-19 [9]. A common feature in adults with severe COVID-19 is regional thrombosis in pulmonary vasculature [15]. The outcome of COVID-19 in adults with PH is worse, although the incidence of SARS-CoV-2 infection in PH patients is similar to the general population [16]. Old age with comorbidities confers a higher risk for death due to COVID-19, but children with PH are not immune to complications of SARS-CoV-2 infection.

## 4. Management of Preexisting or New-Onset PH Occurring with COVID-19

The management of PH occurring with COVID-19 remains primarily supportive care at this time. Supplemental oxygen should be used if oxygen saturation drops below 92% with SARS-CoV-2 infection. The PH medications should not be interrupted during the pandemic in preexisting PH patients. Inhaled nitric oxide (iNO) is known to be beneficial for treating ARDS and persistent PH in newborns. It has been reported that iNO may be useful in pregnant women with severe COVID-19 to prevent vertical transmission of SARS-CoV-2. According to the Chinese expert consensus statement’s recommendation on perinatal and neonatal management for the prevention and control of COVID-19 in newborn respiratory care unit with severe ARDS, high-dose pulmonary surfactant, iNO, and high-frequency oscillatory ventilation may be effective [17]. As the iNO also inhibits SARS-CoV-2 replication [18], this may prevent the progression of the COVID-19 disease. Phosphodiesterase-5 inhibitors (PDEI), especially sildenafil, may counteract the angiotensin type I (AT-1) receptor and reduce pro-inflammatory cytokines and alveolar infiltration of inflammatory cells [19]. Also, sildenafil and tadalafil inhibit the transition of the endothelial and smooth muscle cell to mesenchymal cells in the pulmonary artery, preventing clotting and thrombotic complications. Thus, PDEI helps improve gas exchange. Endothelin-1 is a potent endogenous vasoconstrictor, mainly secreted by endothelial cells. It increases pulmonary vascular tone and induces chronic inflammatory status by producing cytokines, growth factors, collagen, and aldosterone production [20]. Hence, endothelin receptor antagonists (ERAs) can prevent COVID-19-associated lung injury due to its anti-inflammatory actions [21,22].

Hemodynamic profiles in COVID-19 patients with ARDS are consistent with the elevated diastolic pulmonary gradient (the difference between the diastolic pulmonary artery pressure and the pulmonary artery wedge pressure) of >7 and pulmonary vascular resistance (PVR) >3 Wood units x m^2^, suggesting a pre-capillary type of PH [23,24]. RV dysfunction by elevated PVR is a poor prognostic factor in patients with COVID-19 [25]. Right ventricular decompensation can occur rapidly in patients with preexisting PH [26]. There are ongoing trials to explore the role of specific therapies for PH, such as ambrisentan, sildenafil, iloprost, iNO, recombinant ACE2, vasoactive intestinal peptide (VIP) analog, and tocilizumab in COVID-19 [27].

For severe respiratory infections that require admission to hospitals, patients should be admitted to hospitals with intensive care capabilities. Intubation should be avoided if possible, using high-flow nasal cannula oxygen in the prone position. If the patient requires intubation, it should be performed by an experienced physician, and the pre-intubation period is the most critical time as hypotension and hemodynamic collapse can occur. When intubation is planned, vasopressors should be started before induction with anesthesia to prevent sudden collapse due to hypotension. Inhaled NO should be tried as rescue therapy for COVID-19-induced ARDS and refractory hypoxemia before extra-corporeal support [28].

A general approach to patients with PH in children during the COVID-19 pandemic is proposed and illustrated in Figure 2. During the pandemic, any child with existing PH or with newly diagnosed PH should be tested for SARS-CoV-2 if there are signs of fever, respiratory distress, or hypoxemia. If the SARS-CoV-2 antigen test is negative, but clinical COVID-19 is suspected, then antibody screening is recommended [29]. A thorough clinical history including exposure to COVID-19, physical examination, and laboratory tests including chest-X-ray, electrocardiogram, complete blood count, C-reactive protein, D-dimer, lactate dehydrogenase, procalcitonin, brain natriuretic peptide, troponin, blood gas analysis, and an echocardiogram should be performed as early as possible [30,31,32,33]. Early imaging studies, especially CT lungs, are recommended as CT images’ yield is much higher than those of chest-X-ray. However, the pattern of CT changes in the pediatric population is quite different from adults. A typical CT finding observed in children with COVID-19 is lung consolidation with the surrounding halo sign [34]. Quantitative high-resolution CT (HRCT) in COVID-19 patients helps measure pulmonary blood volume changes and can display striking anomalies consistent with blood vessels’ pruning and help assess pulmonary vascular dysfunction. The HRCT findings are correlated well with increased pulmonary vascular resistance [35].

A multidisciplinary team, including an intensivist, pulmonologist, cardiologist, infectious disease specialist, and rheumatologist should be consulted to manage patients with PH and COVID-19 [32]. If there are signs and symptoms of heart failure or evidence of RV dysfunction by cardiac magnetic resonance imaging (CMR), patients should be treated according to guideline-directed medical therapy [36]. No therapeutics have yet been proven effective for the treatment of severe illness caused by COVID-19. European Society of Cardiology guidelines recommend low-dose steroids for children with severe COVID-19-related illness, such as refractory shock, ARDS, and mechanically ventilated patients [37]. In hospitalized patients with severe COVID-19, remdesivir is superior to placebo in shortening recovery time [38]. A subgroup of patients with severe COVID-19 may have a cytokine storm syndrome, and treatment of hyperinflammation using existing therapies such as IL-6 inhibitors (tocilizumab and sarilumab), and interleukin 1 receptor antagonists (IL-1RA) (anakinra) may reduce the mortality [39].

## 5. Impact of COVID-19 on PH Care in Children

COVID-19 has had a significant impact on all aspects of PH care in children and adults, from diagnosis to management. An enormous impact on clinic operations is observed, including fewer clinic visits and significantly increased telemedicine use. Before the COVID-19 pandemic, <10% of PH centers in the United States provided virtual visits, but now 97% of PH centers offer routine virtual visits [3]. The data from international PH registries (Australian and New Zealand [40], the United Kingdom [41], and Germany [42]) suggest that during the SARS-CoV-2 pandemic, there is a delay in referral and a subsequent impact on outcomes in children and adults with PH. While the COVID-19 pandemic is unfolding, the effect of co-infection of SARS-CoV-2 and influenza (COVI-Flu) will soon be devastating. The PH programs have been adopted to establish their protocols for the management of PH during the pandemic without increasing the risk of exposure to or transmission of SARS-CoV-2. Some PH centers have decreased the number of routine diagnostic testing procedures as a part of virtual visits. Only a minority of centers use any experimental medical therapy for COVID-19 [16]. The pandemic has had a second surge in most parts of the world during winter months. Co-infection with influenza or respiratory syncytial virus and COVID-19 at the same time could overwhelm the health care system further and potentially reduce our ability to catch and treat both respiratory illnesses effectively. It is essential to implement mask-wearing, hand sanitization, and social distancing as per CDC guidelines to effectively prevent transmission of SARS-CoV-2 infection and contain the COVID-19 pandemic. The preliminary data on highly effective vaccines against COVID-19 is a hopeful signal that help is on the way.

## 6. Conclusions

The impact of the COVID-19 pandemic and its aftermath will be observed for years to come, as the long-term effect of SARS-CoV-2 infection is unknown. The PH community is adjusting in real time with typical clinical scenarios in adults with COVID-19 and using the best clinical practice in children with PH. It is essential to continue the specific PH therapy in children even during COVID-19 illness unless changed by their provider. Sildenafil and bosentan, commonly used therapies for PH, have shown beneficial effects by shifting the blood flow to adequately ventilated lungs and their anti-inflammatory actions. According to the CDC, there is no evidence to suggest that patients with PH are at higher risk of becoming infected with SARS-CoV-2. However, strong evidence shows that PH can be worsened with pneumonia or other infectious complications due to COVID-19. The worsening of PH symptoms in an established PH patient may suggest a warning sign during the COVID-19 pandemic and necessitates an investigation of SARS-CoV-2 infection. The long-term effects of COVID-19 are unknown, and clinicians gather information through patient care to understand the impact of COVID-19 on PH patients. Pulmonary hypertension centers and international registries continue to monitor the PH population during the COVID-19 to manage PH patients better in the future.

## Figures and Tables

**Figure 1 medicina-56-00716-f001:**
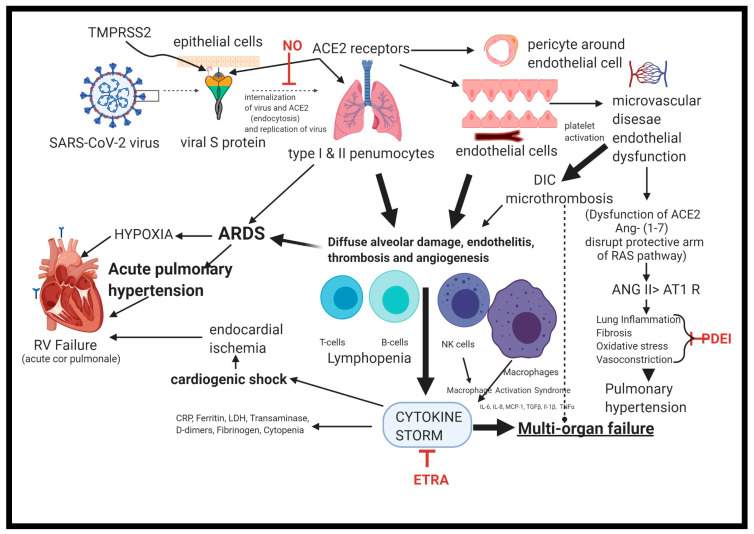
Several putative mechanisms of development of pulmonary hypertension (PH) with COVID-19 and the possible role of PH-specific therapies. (Ang: Angiotensin; AT1 R: Angiotensin 1 receptor; ACE2: Angiotensin-Converting Enzyme; ARDS: acute respiratory distress syndrome; DIC: Disseminated intravascular coagulation; ETRA: Endothelin Receptor Antagonist; IL: Interleukin; LDH: Lactate dehydrogenase; NO: Nitric Oxide; PDEI: Phosphodiesterase Inhibitor; RAS: Renin-Angiotensin-System; RV: Right Ventricle; TMPRSS2: Transmembrane Serine Protease 2).

**Figure 2 medicina-56-00716-f002:**
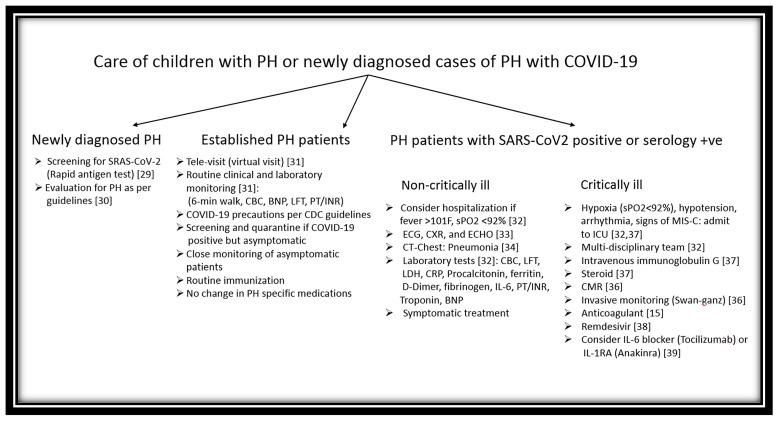
A proposed approach to children with preexisting PH or newly diagnosed PH during the COVID-19 pandemic.

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
