# Peer review of "COVID-19 and Pulmonary Hypertension in Children: What Do We Know So Far?"

_medicina, 2020, doi:10.3390/medicina56120716_

Round 1

Reviewer 1 Report

In this review, the author provides a summary of what is known so far about pulmonary hypertension and COVID-19 in children.

Comments

  1. In the pathophysiology section, the author also discusses treatment and therapy which should be included in a separate section, possibly in the section-Management of PH with COVID-19.
  2. The author should add a section discussing potential explanations for the difference between children and adults in COVID-19.
  3. The author should also include a section to compare the incidence of pulmonary hypertension in children vs adults.
  4. There is a mention of Figure-1 and Figure-2 which is not available otherwise with the manuscript.
  5. The sections, ‘Impact of COVID-19 on PH in Children’ and the ‘Conclusion’ should be more detailed and elaborative.
  6. The author should also do a thorough checkup of the errors in the English language, for example, lines 60-61, “The association between COVID-19 and chronic thromboembolic 59 pulmonary hypertension (CTEPH) development is not common in children as it is too early to comment as the COVID-19 pandemic is still unveiling” and lines 129-130, “There is an urgent need for all children should receive Flu-shot”.

Author Response

  1. In the pathophysiology section, the author also discusses treatment and therapy, which should be included in a separate section, possibly in the section-Management of PH with COVID-19.
  • Thank you for your comment. I separated the treatment section.
  1. The author should add a section discussing potential explanations for the difference between children and adults in COVID-19.
  • A separate section on the difference between children and adults with COVID-19
  1. The author should also include a section to compare the incidence of pulmonary hypertension in children vs. adults.
  • As there is no large scale data available in children, it is not easy to have a separate section on incidence but is incorporated with discussion.
  1. There is a mention of Figure-1 and Figure-2, which is not available otherwise with the manuscript.
  • I apologize as somehow, the version has not been updated online. The figures are added now.
  1. The sections, ‘Impact of COVID-19 on PH in Children’ and the ‘Conclusion’ should be more detailed and elaborative.
  • Both sections are revised.
  1. The author should also do a thorough checkup of the errors in the English language, for example, lines 60-61, “The association between COVID-19 and chronic thromboembolic 59 pulmonary hypertension (CTEPH) development is not common in children as it is too early to comment as the COVID-19 pandemic is still unveiling” and lines 129-130, “There is an urgent need for all children should receive Flu-shot”.
  • I hope the language is corrected in the revised version.

Reviewer 2 Report

Reviewer’s comments:

The reviewer has read the review on COVID and children with PH. The paper is well-organized and well written. It focuses on a small but potentially vulnerable population where large-scale evidence-based medicine will be difficult.

The Reviewer raises a few concerns:

  1. It is a narrative review which should be stated.
  2. Two cases: probably true, but did the author not find all? A systematic search strategy might be doable due to expected small numbers. Systematic review process would increase the scientific weight of the paper
  3. Section 3 (first part): children or all PH patients? Should be clarified.
  4. The recommendations (therapeutic, diagnostic approaches etc) – are they the author’s opinion of a strategy or actual guidelines/recommendations, should be addressed. Many actions “should” be done e.g. giving Flu-shots, consulting multidisciplinary teams, intubation etc. No references appear on these statements.

Minor:

  1. Reference on statement that Covid positive children were with chronic lung disease (line 22-23)?
  2. Justify (possibly correct) statement on PH patients being vulnerable (line 24) with either reference or biomedical explanation.
  3. Stronger division between pathophysiology and treatment? Several therapeutic strategies are mentioned in section 2 and might be more appropriate in Section 3?
  4. Large parts of section 3 is not supported by references. Is this a supportive recommendation by the author?
  5. Check for few spelling mistakes/grammas/sentence constructions (lines 96, 83, 99-102,)
  6. Consider the many unexplained abbreviations (especially line 102-103)
  7. Check and update all references, e.g. reference 26+35+36 miss in the text?
  8. Conclusion should not contain new information not discussed in the main text (face mask etc)

Author Response

  1. It is a narrative review which should be stated.
  • I added that it is a narrative review and added the use of electronic database for the sources.

2. Two cases: probably true, but did the author not find all? A systematic search strategy might be doable due to expected small numbers. A systematic review process would increase the scientific weight of the paper

  • I did a thorough search in Pubmed, Scopus again, and those published in advance (medRxiv and bioRxV) did not find any new cases in children with existing PH with COVID-19.

3. Section 3 (first part): children or all PH patients? Should be clarified.

  • Section 3 is revised. Mostly this applies to children as the data specific for children with PH is not available.

4.The recommendations (therapeutic, diagnostic approaches etc.) – are they the author's opinion of a strategy or actual guidelines/recommendations, should be addressed. Many actions "should" be done e.g. giving Flu-shots, consulting multidisciplinary teams, intubation etc. No references appear on these statements.

  • Thank you. I clarified and described it as a proposed approach.

Minor:

  1. Reference on statement that Covid positive children were with chronic lung disease (line 22-23)?
  • Reference has been updated
  1. Justify (possibly correct) statement on PH patients being vulnerable (line 24) with either reference or biomedical explanation.
  • This has been corrected and revised.
  1. Stronger division between pathophysiology and treatment? Several therapeutic strategies are mentioned in section 2 and might be more appropriate in Section 3?
  • Thank you. The pathophysiology and treatment section is separated and revised.
  1. Large parts of section 3 is not supported by references. Is this a supportive recommendation by the author?
  • All the references in Figure-2 are cited.
  1. Check for few spelling mistakes/grammas/sentence constructions (lines 96, 83, 99-102,)
  • Thank you. I revised the errors.
  1. Consider the many unexplained abbreviations (especially line 102-103)
  • All abbreviations are spelled out fully.
  1. Check and update all references, e.g., reference 26+35+36 miss in the text?
  • I apologize, these references are cited within Figure-2 and as continuation with text, unfortunately, the journal did not incorporate this previously, which are now included.
  1. Conclusion should not contain new information not discussed in the main text (face mask etc)
  • The conclusion is revised.

Round 2

Reviewer 1 Report

In the review entitled, ‘COVID-19 and Pulmonary Hypertension in Children: 2 What do We Know so far?’ the author aims to discuss the pathophysiology of pulmonary hypertension occurring with COVID-19, differences between children and adults with COVID-19, strategies for managing PH in children during the pandemic, and the impact of COVID-4 19 within the field of PH.

However, the author still fails to elaborate on the conclusion as asked in the previous review. In fact, the conclusion section has been shortened from what it was in the earlier version. It is difficult for the reader to draw any serious conclusion from this section. For example, lines 32-33 are inconclusive and not backed by concrete data.

The subsection ‘Pathophysiology of PH occurring with COVID-19’ mainly discusses the pathophysiology of covid-19 and whereas the aim should be to discuss PH in the light of covid-19, which is clearly still missing.
In the management section, the author does not discuss the inhaled nitric oxide therapy (iNO) which is well established in persistent pulmonary hypertension of the newborn (PPHN). Pulmonary hypertension occurs as a complication of severe pneumonia and there are reports of it complicating severe COVID-19 pneumonia.
The article still contains many grammatical errors and needs an extensive revision for the use of the English language and style.

The quality of the figure which appears now is poor and inconsistent in its design.

Author Response

However, the author still fails to elaborate on the conclusion as asked in the previous review. In fact, the conclusion section has been shortened from what it was in the earlier version. It is difficult for the reader to draw any serious conclusion from this section. For example, lines 32-33 are inconclusive and not backed by concrete data.

  • Thank you. The conclusion section is revised and elaborated. Lines 32-33 is the summary of the whole article. Ref#3 (Ryan JJ et al), Ref #31 (Kache S et al), and Ref#36 (ESC Guidelines) all suggest that the most important is to continuity and adherence to concurrent pulmonary hypertension therapy. Ref#18 (Isidori AM et al) describes the role of Sildenafil. Ref #21 (Badagliacca R wt al) and Ref #30 suggest role of bosentan. and Ref #26 describes all the current trials with these medications also.

The subsection ‘Pathophysiology of PH occurring with COVID-19’ mainly discusses the pathophysiology of covid-19 and whereas the aim should be to discuss PH in the light of covid-19, which is clearly still missing. 

  • The pathophysiology section is revised and more information of mechanism of pulmonary hypertension.

In the management section, the author does not discuss the inhaled nitric oxide therapy (iNO) which is well established in persistent pulmonary hypertension of the newborn (PPHN). Pulmonary hypertension occurs as a complication of severe pneumonia and there are reports of it complicating severe COVID-19 pneumonia.

  • Thank you. Added role of iNO in PPHN babies born to COVID-19 positive mother. PH as a complication of severe pneumonia is added.

The article still contains many grammatical errors and needs an extensive revision for the use of the English language and style.

  • Thank you. I have used Grammarley expertise to check the grammar and spelling. If there are still mistakes, I will use MDPI language service.

The quality of the figure which appears now is poor and inconsistent in its design.

  • The Figure-1 and 2 are prepared in biorender.com professional service and the pictures are supplied as scientifically appropriate. I would appreciate if the reviewer could pointout the inconsistencies, will consult biorender.com to fix these.